# Improvement of Ex Vitro Growing Completion of Highbush Blueberry (*Vaccinium corymbosum* L.) in Containers

Svetlana Akimova [1,*], Agamagomed Radzhabov [1,*], Aleksandr Esaulko [2], Egor Samoshenkov [1], Ivan Nechiporenko [1,3,*], Pavel Kazakov [1,3], Yurii Voskoboinikov [1], Anna Matsneva [1], Aleksandr Zubkov [1] and Timur Aisanov [2]

[1] Russian State Agrarian University—Moscow Timiryazev Agricultural Academy, Institute of Horticulture and Landscape Architecture, St. Timiryazevskaya, 49, 127434 Moscow, Russia

[2] Faculty of Agrobiology and Land Resources, Ecology and Landscape Architecture, Stavropol State Agrarian University, 12, Zootechnichesky Lane, 355017 Stavropol, Russia

[3] All-Russian Phytopathology Research Institute, St. Institute, 143050 Big Vyazyomy, Russia

\* Correspondence: akimova@rgau-msha.ru (S.A.); plod@rgau-msha.ru (A.R.); vannechiporenko@gmail.com (I.N.)

**Abstract:** Highbush blueberry (*Vaccinium corymbosum* L.), originally a forest plant, is currently in need of improvement of clonal micropropagation technologies. It is known that the large percentage of propagated plants can be lost or damaged, not only at the stage of acclimatization to non-sterile conditions, but also during the growing completion stage. In fact, successful ex vitro regeneration of such plants is determined by their ability to produce new shoots that can adapt to new cultivation conditions. The lighting and ratio of nutrients under ex vitro conditions play an important role in the development of the plants' photosynthetic capacity. The research revealed that LED grow lighting has a positive effect on the development of ex vitro plants of highbush blueberry (*Vaccinium corymbosum* L.) cv. Brigitta Blue, only at the initial stages of growing in 0.5-L containers. The results obtained have improved our understanding of lighting and mineral fertilizer's impact on the development of ex vitro plants of the highbush blueberry (*Vaccinium corymbosum* L.) cv. Brigitta Blue in greenhouse conditions. This can be useful for providing blueberry planting stock and commercial use for large scale production.

**Keywords:** highbush blueberry; vegetative propagation; clonal micropropagation; growing ex vitro plants; of growing completion; growlights; Biogeosystem Technique

## 1. Introduction

Shrubs are an important part of the forest ecosystem. Similar to any other shrub of the species *Ericacea*, highbush blueberry (*Vaccinium corymbosum* L.) originates from North American forest and plays an important role in modern horticulture. The highbush blueberry is a valuable berry plant, and measures are required to improve highbush blueberry plant propagation and prepare the soil in the plantations to ensure heavy metal (HM) free production [1–3].

The highbush blueberry (*Vaccinium corymbosum* L.) is a species of the genus *Vaccinium* L., family *Ericaceae* Juss, section *Cyanococcus* and is considered a promising berry crop worldwide, both economically and biologically. It has high nutritional value, due to significant amount of vitamins and biologically active substances [4,5].

According to the Blueberry International Organization, for the period 2016–2020 blueberry cultivation area in the world expanded from 132.56 to 205.67 thousand hectares, which means an increase of 73.1 thousand hectares, mainly due to expansion of cultivation areas in China, Peru, Poland and other regions. World blueberry production in 2020 amounted to over 850 thousand tons. At the same time, North America is the leader in

highbush blueberry production (57% of world yield), followed by South America (23%), Europe (11%) and the Asian–Pacific region (8%) [6,7].

In the Russian Federation, according to its agroclimatic characteristics, this valuable berry crop is considered suitable for growing in many regions, but only the very first steps have been taken to introduce it into industrial production. Due to specific requirements for soil and climatic conditions, its introduction into cultivation is associated with additional research that adjusts the methods of vegetative propagation and cultivation techniques [8].

Presently, traditional vegetative propagation of highbush blueberry (*Vaccinium corymbosum* L.) by lignified and green cuttings is experiencing difficulties due to poor rooting, a need for numerous mother plants, and limited seasonal shoot growth. The crop is not propagated by seed, since the seedlings do not retain biological and economic characteristics as a result of cross-pollination [5].

Recently, to obtain the required amount of high-quality planting material of the genus *Vaccinium* L., a clonal micropropagation technique has been proven efficient. This is a modern and intensive method of agamic plant mass reproduction using tissue culture [9,10]. This technique also enables us to redeem regenerated plant tissues from many pathogens that reduce vegetative development and plant productivity. Plant body rejuvenation after in vitro culture enhances the plants' capacity for vegetative propagation, vigor and productivity [11–20].

A large number of present-day studies in the field of clonal micropropagation are focused on laboratory experiments. However, the information on how plants behave after completing their growth stage appears insufficient. During the ex vitro stage, severe stress is often noted that can cause death or stunted growth of the ex vitro plants under non-sterile conditions [21–23].

Currently, one of the problems of the clonal micropropagation technique is that a large percentage of propagated plants can be lost or damaged, not only during adjustment to non-sterile conditions, but also at the stage of growing completion [24–27]. Plants are known to develop in vitro under conditions characterized by high humidity, stable nutrition, free from external infection, controlled temperature and photoperiod. As a result of heterotrophic nutrition of plants in vitro, their leaf apparatus loses its ability for active photosynthesis partially or fully, and roots of microplants are often devoid of root hair, which is associated with the lack of oxygen, which hinders water absorption and mineral nutrition [15,22,28–31].

The adaptation period includincludedes at least four simultaneous processes: adaptation of the assimilating apparatus to low air humidity and new infection load, adaptation of adventitious roots to the substrate and soil microbiome. The main objective is to achieve the functionality of the root system while maintaining air humidity close to 100% for the aerial part with a relative sterility of the substrate. The chief criterion for the microplant's survival is the beginning of the growth of the above-ground system, which indicates the adaptation of the root system to the conditions of the new substrate, most often taking 2–3 weeks [32–34].

Plants' ex vitro growing completion under greenhouse conditions in containers has a number of advantages. Firstly, survival rate increases when the plant is transplanted into open ground conditions, the transplantation time and the time to obtain standard seedlings are reduced. Secondly, when planting material with a closed root system in a permanent place throughout the growing season, labor costs during transportation and storage are significantly reduced. Thirdly, protected cultivation controls pests, diseases, and weeds, so the amount of applied pesticide is therefore reduced [28,35–37].

One of the key points of acclimatization and subsequent adaptation should be the resumption of the functioning of the stomatal apparatus, since in vitro the plant is in conditions of high humidity, stable nutrition, and free from external infection, which could slow down metabolic processes [22,38].

In the experiments of Hung C.D. and others, advantages of in vitro cultivation at the stages of multiplication and rhizogenesis has been demonstrated, particularly, for

*Musa* sp., *Fragaria* cv. Akihime, *Fragaria x ananassa* Duch. cv. Camarosa, *Euphobia millii*, *Tripterospermum japonicum*, *Vanilla planifolia* Andrews, *Stevia rebaudiana* Betroni var. Morita II, *Vaccinium ashei* Reade cv. Titan, *Vaccinium corymbosum* L. cv. Huron under LED vs. fluorescent, and efficient completing of growth ex vitro under LED lighting [39,40].

It is known that highbush blueberry (*Vaccinium corymbosum* L.) has a specific shallow root system, lacking root hair and located in the upper soil layer, where the plant is nourished by endotrophic mycorrhiza. In order for the plant to grow and develop well, it is necessary to create conditions that stimulate the development of mycorrhizal fungi and optimize the conditions for the development of perennial plantations [41,42]. Ex vitro highbush blueberry (*Vaccinium corymbosum* L.) plants are characterized by very slow growth and development, which can be caused by various reasons, such as a lack of nutrients, insufficient lighting, incorrect selection of a light spectrum and temperature conditions, which may causes leaf burning, poor growth and shoot wilting [43,44].

It is also important to use a particular substrate and containers for further growing ex vitro plants. High-moor (sphagnum) peat is considered to be the best one for blueberries, due to its water content and air capacity for the root system, as well as the necessary acidity of the substrate, which ensures vital activity of plants [45,46].

In addition, under ex vitro conditions, leaf activity can be violated due to photoinhibition, found to develop under in vitro conditions and associated with the stress experienced by the leaf apparatus when cultivation conditions are suddenly and markedly changed [22,45]. The spectral composition of light also plays an important role. In addition to stimulating photosynthetic processes, it can positively influence morphometric indicators of plant development and secondary metabolite production. Numerous studies have confirmed that the spectral combination of red and blue light in various ratios is quite effective for crop growing in greenhouse conditions [47]. The best photosynthetic active response in ex vitro plants is observed with white and red-blue light spectrum [48–50].

Blue light is effective for inducing and accumulating chlorophyll content, carbonnitrogen ratio (C/N ratio) and plastid number of guard cells in ex vitro *Actinidia* plants, while red light induces vegetative growth [51]. Therefore, it appears promising to select the optimal levels of illumination and mineral nutrition regimes that will help to avoid problems and optimize the highbush blueberry (*Vaccinium corymbosum* L.) growing in ex vitro conditions. At the same time, it is important to overcome the limitations of organogenesis, which are imposed by the use of outdated control technologies, both in the production of planting material and the cultivation of perennial industrial open-ground plantations [52–54].

There is almost no information on how plants' blueberries develop after the commercially valuable stage of clonal micropropagation—'of growing completion', or 'ex vitro post-adaptation of plants' (growing up to marketable state). Many researchers confuse this stage with adaptation and subsequent acclimatization. But what happens to the plants after—before the seedlings are sold in nurseries or planted in plantation?

The aim of our research was to improve the methods of growing ex vitro plants of highbush blueberry (*Vaccinium corymbosum* L.) in greenhouse conditions using two types of lighting (LED lighting with a 16-h photoperiod and natural lighting—without additional lighting devises) and of mineral fertilizers. This will ensure better plant growth in situ.

## 2. Materials and Methods

Experiments were carried out at Russian State Agrarian University—Moscow Timiryazev Agricultural Academy, Department of biotechnology and berry crops of Edelstein Educational Scientific and Production Center for Horticulture and Vegetable Growing in 2021–2022.

Blueberry microplants were obtained by clonal micropropagation. At the stage of multiplication, two sequential passages were produced on the nutritive growth medium based on Woody Plant Medium (WPM) enriched with the following substances (mg/L): thiamine hydrochloride (B1), pyridoxine hydrochloride (B6), nicotinamide (PP)—0.5; meso-inositol—100, sucrose—30,000, with the addition of 2-iP (2-isopentenyladenine) at concentrations

of 2.5 mg/L. After the ingredients were combined, the media was adjusted to pH 4.5 and agar-agar—8 g/L. The cultures in vitro were subcultivated in a light room at a lighting intensity of 500 lux under lighting mixed (grow—PPFD 18.9 $\mu mol/s^{-1}/m^{-2}$ and fluorescent—PPFD 43.0 $\mu mol/s^{-1}/m^{-2}$ lamps), a 16-h photoperiod and the temperature of 20–22 °C. Subcultivation period was 60 days [55,56].

At the stage of rooting and acclimatization, the microplants were planted in seedling cassettes, 144 cells containing peat with agroperlite. They were placed in a humid chamber with a vaporizer (temperature 22 ± 2 °C, humidity 90%, PPFD 1120 $\mu mol/s^{-1}/m^{-2}$, 16 photoperiod) for rooting for 45 days [57]. Rooting rate of the microplants was 90%–95%. Upon adjustment to non-sterile conditions, the plants were maintained in the greenhouse (temperature 24–30 °C, humidity 75%).

Ex vitro plants of highbush blueberry (*Vaccinium corymbosum* L.) cv. Brigitta Blue were the object of the research (Figure 1). This highbush blueberry (*Vaccinium corymbosum* L.) cultivar was obtained by free pollination of cv. Lateblue in Australia in 1980. These plants are upright, well-branched bushes. Depending on the cultivation area, it is considered a mid-late or late cultivar. Berries are medium in size, sweet in taste. Long-term postharvest storage of the berries may last up to 7 weeks. The crop is suitable for industrial plantations [58,59].

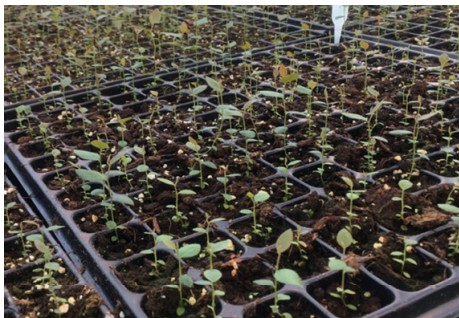
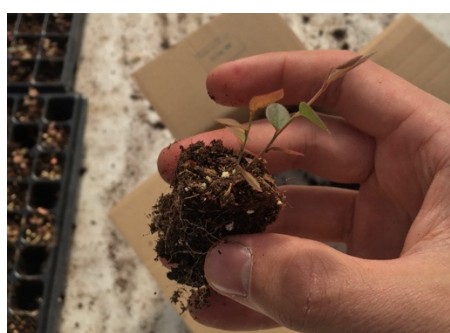

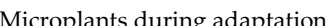
Microplants during adaptation        Rooted microplant

**Figure 1.** Rooting process of highbush blueberry (*Vaccinium corymbosum* L.) cv. Brigitta Blue.

In the first ten days of April, adapted blueberry plants were planted into 0.5-L containers with peat substrate 'Veltorf' of an acidic pH circa 3.5–4; mineral fertilizers were added according to the options: $N_{16}P_{16}K_{16}$ (produced by ACRON) 0.2 and 0.4 g/L, APAVIVA $N_{15}P_{15}K_{15}(S_{10})$ (produced by PhosAgro) 0.2 and 0.4 g/L, control—no fertilizers applied.

The plants were exposed to different types of illumination: in section of growing completion under LED growlights (UnionPowerStar—40W-T) with a photoperiod of 16 h and natural light (without additional illumination tools) in greenhouse conditions.

Grow lamps UnionPowerStar—40W-T have photosynthetic photon flux density (PPDF) 70 $\mu mol/s^{-1}/m^{-2}$ at a distance 50 cm from plants and 386 $\mu mol/s^{-1}/m^{-2}$ at a distance 10 cm from plants in the ratio red and blue light 3:1, which is optimal for various crops in vitro and during the period adaption [60,61]. Under greenhouse conditions PPFD was found in a ratio from 80 to 250 $\mu mol/s^{-1}/m^{-2}$ depending on cloud and during the day times. According to some data, the optimum light intensity for blueberry (*Vaccinium* sp.) plants should be concentrations below 100 $\mu mol/s^{-1}/m^{-2}$, which causes stronger growth [62].

It has been proven, that the better intensity light is causing abatement of leaf blade length, which is compensated by its growth in width, while a disbalance of the volume of top parts and of root system is arising, which can depress growth at time transplanting [61,63]. Therefore, during the period of adaptation and growing completion of plants after in vitro culture, it is important to gradually increase the intensity of illumination, for the best formation of the root system, which is quite difficult to do under natural illumination in a greenhouse. After growth of adapted blueberry (*Vaccinium* sp.) plants, intensive

LED illumination should be increased gradually, without abrupt differentials, reducing plant stress.

The morphometric parameters of ex vitro plant development were recorded four times every fortnight (two weeks) on days 14, 28, 42 and 56. The number of shoots (0th and 1st orders of branching), the total length of shoots and leaf surface areas were counted. After the 4th count, the plants were transplanted into 2 L containers, adding the above stated doses of mineral fertilizers ($N_{16}P_{16}K_{16}$ 0.2 and 0.4 g/L, $N_{15}P_{15}K_{15}(S_{10})$ 0.2 and 0.4 g/L). Development was recorded after 1.5 months—on day 112.

Experiments were performed in triplicate, 25 plants per repetition. Statistical processing of the results was carried out according to the method of A.V. Isachkin, using Microsoft Office Excel 2007 [64].

## 3. Results

Observations revealed that the type of illumination and fertilizer significantly affected the growth and development of ex vitro highbush blueberry (*Vaccinium corymbosum* L.) cv. Brigitta Blue plants, both separately and in mutual interaction. On the 14th day of growing completion, during the first registration of morphometric indicators in ex vitro highbush blueberry cv. Brigitta Blue plants, significant differences were detected between the controls and experimental plants exposed to growlight (UnionPowerStar—40W-T) as well as plants to which fertilizers were applied (options: $N_{16}P_{16}K_{16}$ concentration of 0.4 g/L and $N_{15}P_{15}K_{15}(S_{10})$ concentration 0.2 g/L). Average total shoot length was $12.7 \pm 1.80$ and $16.4 \pm 6.73$ cm versus $8.5 \pm 1.35$ cm in the controls (no fertilizers), and the leaf surface area was $7.9 \pm 2.15$ and $13.0 \pm 5.80$ cm$^2$ versus $3.6 \pm 1.11$ cm$^2$ in the controls.

Under natural light conditions, a significant difference in the average number of the 1st order shoots were found only in the plants with $N_{15}P_{15}K_{15}(S_{10})$ applied (dosage 0.4 g/L), where this number was $3.2 \pm 1.6$ pcs. versus 0 pcs. in the controls (Table 1, Figure 2).

**Table 1.** Morphometric indicators of ex vitro development in highbush blueberry plants cv. Brigitta Blue on the 14th day of growing completion.

| Type of Fertilizer (Factor B) | Illumination (Factor A) | | Factor Average B |
|---|---|---|---|
| | Natural Lighting + SD * | Light-Emitting Diode (LED) + SD | |
| Average number of the 0th order, pcs. | | | $LSD_{05}$ b = $F_e < F_t$ *** |
| no fertilizers (control) | $3.0 \pm 1.41$ | $2.2 \pm 0.45$ | 2.6 |
| $N_{16}P_{16}K_{16}$ 0.2 g/L | $2.4 \pm 0.55$ | $2.4 \pm 0.55$ | 2.4 |
| $N_{16}P_{16}K_{16}$ 0.4 g/L | $2.8 \pm 0.84$ | $3.2 \pm 0.84$ | 3.0 |
| $N_{15}P_{15}K_{15}(S_{10})$ 0.2 g/L | $2.8 \pm 0.45$ | $3.2 \pm 1.10$ | 3.0 |
| $N_{15}P_{15}K_{15}(S_{10})$ 0.4 g/L | $3.0 \pm 0.71$ | $2.8 \pm 0.84$ | 2.9 |
| Factor average A $LSD_{05}$ a = $F_e < F_t$ | 2.8 | 2.8 | |
| $LSD_{05}$ ab = $F_e < F_t$ | | | |
| Average number of the 1st order, pcs. | | | $LSD_{05}$ b = $F_e < F_t$ |
| no fertilizers (control) | 0 | $0.2 \pm 0.00$ | 0.1 |
| $N_{16}P_{16}K_{16}$ 0.2 g/L | $0.6 \pm 0.71$ | $0.8 \pm 0.58$ | 0.7 |
| $N_{16}P_{16}K_{16}$ 0.4 g/L | $0.8 \pm 0.00$ | $1.2 \pm 0.58$ | 1.0 |
| $N_{15}P_{15}K_{15}(S_{10})$ 0.2 g/L | 0 | $1.2 \pm 2.83$ | 0.6 |
| $N_{15}P_{15}K_{15}(S_{10})$ 0.4 g/L | $3.2 \pm 2.16$ [ab, **] | $0.4 \pm 0.00$ | 1.8 |
| Factor average A $LSD_{05}$ a = $F_e < F_t$ | 1.5 | 0.8 | |
| $LSD_{05}$ ab = 2.8 | | | |

**Table 1.** *Cont.*

| Type of Fertilizer (Factor B) | Illumination (Factor A) | | Factor Average B |
|---|---|---|---|
| | Natural Lighting + SD * | Light-Emitting Diode (LED) + SD | |
| The total length of the shoots, cm | | | LSD$_{05}$ b = F$_e$ < F$_t$ |
| no fertilizers (control) | 11.8 ± 4.93 | 8.5 ± 1.35 | 10.15 |
| N$_{16}$P$_{16}$K$_{16}$ 0.2 g/L | 11.3 ± 1.52 | 8.3 ± 1.00 | 9.8 |
| N$_{16}$P$_{16}$K$_{16}$ 0.4 g/L | 11.9 ± 2.01 | 12.7 ± 1.80 [b] | 12.3 |
| N$_{15}$P$_{15}$K$_{15}$(S$_{10}$) 0.2 g/L | 11.2 ± 1.71 | 16.4 ± 6.73 [b, ab] | 13.8 |
| N$_{15}$P$_{15}$K$_{15}$(S$_{10}$) 0.4 g/L | 13.9 ± 0.63 | 13.0 ± 2.31 [b] | 13.5 |
| Factor average A LSD$_{05}$ a = F$_e$ < F$_t$ | 12.0 | 11.8 | |
| LSD$_{05}$ ab = 6.3 | | | |
| Leaf surface area, cm$^2$ | | | LSD$_{05}$ b = 3.7 |
| no fertilizers (control) | 9.2 ± 5.53 | 3.6 ± 1.11 | 6.4 |
| N$_{16}$P$_{16}$K$_{16}$ 0.2 g/L | 5.6 ± 1.60 | 5.0 ± 1.79 | 5.3 |
| N$_{16}$P$_{16}$K$_{16}$ 0.4 g/L | 5.1 ± 1.68 | 7.9 ± 2.15 [b] | 6.5 |
| N$_{15}$P$_{15}$K$_{15}$(S$_{10}$) 0.2 g/L | 4.8 ± 0.55 | 13.0 ± 5.80 [b, ab] | 8.9 |
| N$_{15}$P$_{15}$K$_{15}$(S$_{10}$) 0.4 g/L | 3.8 ± 2.01 | 5.2 ± 2.02 | 4.5 |
| Factor average A LSD$_{05}$ a = F$_e$ < F$_t$ | 5.7 | 6.9 | |
| LSD$_{05}$ ab = 6.2 | | | |

The least significant difference $p < 0.05$ was calculated by two-way variance analysis; * the results were expressed as mean ± standard deviation (SD); ** "a, b, ab"—the difference between the average with the control is significant, based on the comparison of the differences between the average with LSD at a 5% significance level: "a"—by factor "a" (illumination), "b"—by factor "b" (type of fertilizer), "ab"—in the combination of factors; *** "F$_e$ < F$_t$"–F empirical < F theoretical, not proven difference between averages with LSD at 5% significance level.

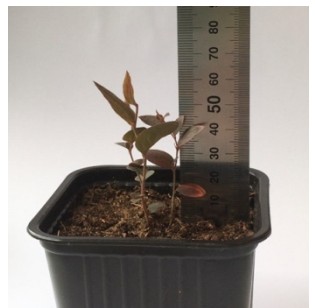 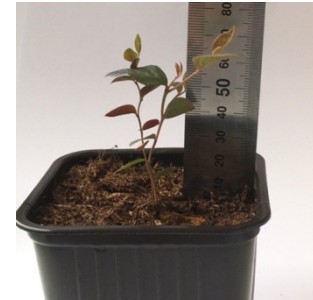 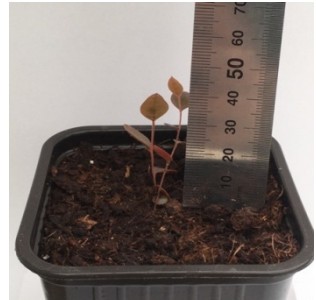 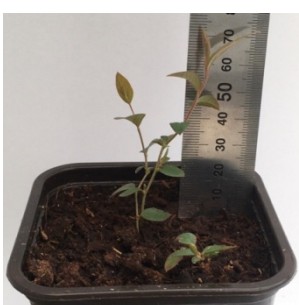

| no fertilizers (control) | N$_{15}$P$_{15}$K$_{15}$ (S$_{10}$) 0.2 g/L | no fertilizers (control) | N$_{15}$P$_{15}$K$_{15}$(S$_{10}$) 0.2 g/L |
|---|---|---|---|
| Light-emitting diode (LED) | | Natural lighting | |

**Figure 2.** External appearance of ex vitro highbush blueberry (*Vaccinium corymbosum* L.) plants on the 14th day of growing completion.

During the second recording on the 28th day of observations, there was an advance of the previously identified variants to LED growlights (UnionPowerStar—40W-T)—N$_{16}$P$_{16}$K$_{16}$ at a concentration of 0.4 g/L and N$_{15}$P$_{15}$K$_{15}$(S$_{10}$) at concentrations of 0.2 g/L and 0.4 g/L.

In addition, under natural light (factor "a"), all experimental variants significantly affected the number of the 1st order shoots (2.8 ± 2.87–3.2 ± 1.64 pcs. vs. 0 ± 0.00 pcs. in the control). A significant effect of the fertilizer type (factor "b") in the N$_{15}$P$_{15}$K$_{15}$(S$_{10}$) variant (fertilizers concentration—0.4 g/L) on the total length of shoots (23.4 ± 5.53 cm versus 14.5 ± 7.17 cm in the control), as well as a leaf surface area (26.9 ± 17.40 cm$^2$, against 11.3 ± 4.55 cm$^2$ in the control) was also revealed (Table 2).

**Table 2.** Morphometric indicators of ex vitro development in highbush blueberry (*Vaccinium corymbosum* L.) plants cv. Brigitta Blue on the 28th day of growing completion.

| Type of Fertilizer (Factor B) | Illumination (Factor A) | | Factor Average B |
|---|---|---|---|
| | **Natural Lighting + SD *** | **Light-Emitting Diode (LED) + SD** | |
| | Average number of the 0th order, pcs. | | $LSD_{05}$ b = $F_e < F_t$ *** |
| no fertilizers (control) | $3.0 \pm 1.41$ | $2.4 \pm 0.55$ | 2.7 |
| $N_{16}P_{16}K_{16}$ 0.2 g/L | $2.6 \pm 0.55$ | $2.6 \pm 0.55$ | 2.6 |
| $N_{16}P_{16}K_{16}$ 0.4 g/L | $2.8 \pm 0.84$ | $3.2 \pm 0.84$ | 3.0 |
| $N_{15}P_{15}K_{15}(S_{10})$ 0.2 g/L | $2.8 \pm 0.55$ | $3.2 \pm 1.22$ | 3.0 |
| $N_{15}P_{15}K_{15}(S_{10})$ 0.4 g/L | $3.0 \pm 0.84$ | $2.8 \pm 0.84$ | 2.9 |
| Factor average A $LSD_{05}$ a = $F_e < F_t$ | 2.8 | 2.8 | |
| | $LSD_{05}$ ab = $F_e < F_t$ | | |
| | Average number of the 1st order, pcs. | | $LSD_{05}$ b = 1.7 |
| no fertilizers (control) | $0.0 \pm 0.00$ | $1.0 \pm 0.00$ | 0.5 |
| $N_{16}P_{16}K_{16}$ 0.2 g/L | $3.0 \pm 1.58$ [b] ** | $1.7 \pm 0.58$ | 2.6 |
| $N_{16}P_{16}K_{16}$ 0.4 g/L | $2.0 \pm 1.15$ [b] | $2.0 \pm 0.71$ | 2.0 |
| $N_{15}P_{15}K_{15}(S_{10})$ 0.2 g/L | $2.8 \pm 2.87$ [b] | $2.0 \pm 1.22$ | 2.4 |
| $N_{15}P_{15}K_{15}(S_{10})$ 0.4 g/L | $3.2 \pm 1.64$ [b] | $1.5 \pm 0.58$ | 2.6 |
| Factor average A $LSD_{05}$ [a] = $F_e < F_t$ | 2.2 | 1.6 | |
| | $LSD_{05}$ [ab] = 2.8 | | |
| | The total length of the shoots, cm | | $LSD_{05}$ b = 6.7 |
| no fertilizers (control) | $14.5 \pm 7.17$ | $10.4 \pm 0.93$ | 12.5 |
| $N_{16}P_{16}K_{16}$ 0.2 g/L | $20.3 \pm 4.24$ | $16.1 \pm 4.64$ | 18.2 |
| $N_{16}P_{16}K_{16}$ 0.4 g/L | $17.9 \pm 6.01$ | $26.0 \pm 4.09$ [b, ab] | 22.3 |
| $N_{15}P_{15}K_{15}(S_{10})$ 0.2 g/L | $14.7 \pm 2.28$ | $25.0 \pm 9.02$ [b, ab] | 19.9 |
| $N_{15}P_{15}K_{15}(S_{10})$ 0.4 g/L | $23.4 \pm 5.53$ [b] | $18.5 \pm 4.48$ [b] | 21.0 |
| Factor average A $LSD_{05}$ [a] = $F_e < F_t$ | 18.16 | 19.3 | |
| | $LSD_{05}$ [ab] = 11.2 | | |
| | Leaf surface area, $cm^2$ | | $LSD_{05}$ b = 12.0 |
| no fertilizers (control) | $11.3 \pm 4.55$ | $6.6 \pm 2.61$ | 8.9 |
| $N_{16}P_{16}K_{16}$ 0.2 g/L | $21.6 \pm 6.67$ | $18.3 \pm 9.80$ | 20.0 |
| $N_{16}P_{16}K_{16}$ 0.4 g/L | $16.7 \pm 7.22$ | $26.0 \pm 15.29$ [b] | 21.4 |
| $N_{15}P_{15}K_{15}(S_{10})$ 0.2 g/L | $10.3 \pm 2.71$ | $35.5 \pm 10.64$ [b, ab] | 23.0 |
| $N_{15}P_{15}K_{15}(S_{10})$ 0.4 g/L | $26.9 \pm 17.40$ [b] | $20.2 \pm 4.54$ [b] | 23.6 |
| Factor average A $LSD_{05}$ [a] = $F_e < F_t$ | 17.4 | 21.3 | |
| | $LSD_{05}$ [ab] = 19.9 | | |

The least significant difference $p < 0.05$ was calculated by two-way variance analysis; * the results were expressed as mean $\pm$ standard deviation (SD); ** "a, b, ab"—the difference between the average with the control was significant based on the comparison of the differences between the average with LSD at a 5% significance level: "a"—by factor "a" (illumination), "b"—by factor "b" (type of fertilizer), "ab"—in the combination of factors; *** "$F_e < F_t$"–F empirical < F theoretical, not proven difference between averages with LSD at 5% significance level.

During the third recording on the 42nd day of growing completion of highbush blueberry (*Vaccinium corymbosum* L.) plants in containers, advance of plants exposed to LED growlights (UnionPowerStar—40W-T) with $N_{16}P_{16}K_{16}$ at a concentration 0.4 g/L and $N_{15}P_{15}K_{15}(S_{10})$ at a concentration 0.2 g/L. At the same time, the lighting conditions (factor "a") significantly affected only a number of the 1st order of shoots of branching, which was respectively $2.6 \pm 2.51$ and $3.4 \pm 1.34$ pcs compared with $1.0 \pm 0.50$ pcs in the controls without fertilizers. The fertilizer type (factor "b") and both factors in combination (ab) significantly affected the total length of shoots ($37.7 \pm 8.73$ and $45.6 \pm 8.49$ cm versus $15.4 \pm 2.45$ cm in the control) and a leaf surface area ($65.0 \pm 20.60$ and $70.6 \pm 16.87$ $cm^2$ versus $12.2 \pm 3.50$ $cm^2$ in the control).

When growing in a greenhouse under natural light, the advance and predominance of the following variants: $N_{16}P_{16}K_{16}$ at concentrations of 0.2 and 0.4 g/L, $N_{15}P_{15}K_{15}(S_{10})$ at a concentration of 0.4 g/L was revealed, in which a number of the 1st order shoots was $3.8 \pm 5.56$ and $7.2 \pm 1.79$ pcs. compared with $0.6 \pm 0.00$ pcs. in the control. A significant effect of a mineral nutrition (factor "b") and an interaction of factors ("ab") on the total length of shoots ($38.1 \pm 13.12$ and $43.2 \pm 13.79$ cm versus $16.1 \pm 8.43$ cm in the control) and leaf surface area ($49.7 \pm 28.51$ and $60.2 \pm 19.30$ cm$^2$ versus $9.0 \pm 5.66$ cm$^2$ in the control) was also revealed (Table 3).

**Table 3.** Morphometric indicators of ex vitro development in highbush blueberry (*Vaccinium corymbosum* L.) plants cv. Brigitta Blue on the 42nd day of growing completion.

| Type of Fertilizer (Factor B) | Illumination (Factor A) | | Factor Average B |
|---|---|---|---|
| | Natural Lighting + SD * | Light-Emitting Diode (LED) + SD | |
| *Average number of the 0th order, pcs.* | | | $LSD_{05}$ b = $F_e < F_t$ *** |
| no fertilizers (control) | $3.0 \pm 1.41$ | $2.6 \pm 0.55$ | 2.8 |
| $N_{16}P_{16}K_{16}$ 0.2 g/L | $2.6 \pm 0.55$ | $2.6 \pm 0.55$ | 2.6 |
| $N_{16}P_{16}K_{16}$ 0.4 g/L | $3.0 \pm 0.71$ | $3.2 \pm 0.84$ | 3.1 |
| $N_{15}P_{15}K_{15}(S_{10})$ 0.2 g/L | $2.8 \pm 0.55$ | $3.2 \pm 1.22$ | 3.0 |
| $N_{15}P_{15}K_{15}(S_{10})$ 0.4 g/L | $3.0 \pm 0.71$ | $2.8 \pm 0.84$ | 2.9 |
| Factor average A $LSD_{05}$ a = $F_e < F_t$ | 2.9 | 2.9 | |
| | $LSD_{05}$ ab = $F_e < F_t$ | | |
| *Average number of the 1st order, pcs.* | | | $LSD_{05}$ b = 2.8 |
| no fertilizers (control) | $0.6 \pm 0.00$ | $1.0 \pm 0.50$ | 0.8 |
| $N_{16}P_{16}K_{16}$ 0.2 g/L | $4.6 \pm 1.52$ [a, b] ** | $2.2 \pm 1.30$ | 3.4 |
| $N_{16}P_{16}K_{16}$ 0.4 g/L | $3.2 \pm 1.30$ | $3.4 \pm 1.34$ [a] | 3.3 |
| $N_{15}P_{15}K_{15}(S_{10})$ 0.2 g/L | $3.8 \pm 5.56$ [a, b] | $2.6 \pm 2.51$ [a] | 3.2 |
| $N_{15}P_{15}K_{15}(S_{10})$ 0.4 g/L | $7.2 \pm 1.79$ [a, b] | $3.0 \pm 2.00$ [a] | 5.1 |
| Factor average A $LSD_{05}$ a = 1.3 | 3.9 | 2.4 | |
| | $LSD_{05}$ ab = $F_e < F_t$ | | |
| *The total length of the shoots, cm* | | | $LSD_{05}$ b = 24.7 |
| no fertilizers (control) | $16.1 \pm 8.43$ | $15.4 \pm 2.45$ | 15.8 |
| $N_{16}P_{16}K_{16}$ 0.2 g/L | $40.2 \pm 10.42$ [b, ab] | $27.0 \pm 9.26$ | 33.6 |
| $N_{16}P_{16}K_{16}$ 0.4 g/L | $38.1 \pm 13.12$ [b, ab] | $45.6 \pm 8.49$ [b, ab] | 41.9 |
| $N_{15}P_{15}K_{15}(S_{10})$ 0.2 g/L | $26.3 \pm 10.02$ | $37.7 \pm 8.73$ [b, ab] | 32.0 |
| $N_{15}P_{15}K_{15}(S_{10})$ 0.4 g/L | $43.2 \pm 13.79$ [b, ab] | $32.7 \pm 7.90$ [b] | 38.0 |
| Factor average A $LSD_{05}$ a = $F_e < F_t$ | 32.8 | 31.7 | |
| | $LSD_{05}$ ab = 41.0 | | |
| *Leaf surface area, cm$^2$* | | | $LSD_{05}$ b = 24.7 |
| no fertilizers (control) | $9.0 \pm 5.66$ | $12.2 \pm 3.50$ | 10.5 |
| $N_{16}P_{16}K_{16}$ 0.2 g/L | $60.2 \pm 19.30$ [b, ab] | $42.4 \pm 19.45$ [b] | 51.3 |
| $N_{16}P_{16}K_{16}$ 0.4 g/L | $58.9 \pm 34.89$ [b, ab] | $70.6 \pm 16.87$ [b, ab] | 64.8 |
| $N_{15}P_{15}K_{15}(S_{10})$ 0.2 g/L | $22.2 \pm 5.28$ | $65.0 \pm 20.60$ [b, ab] | 43.6 |
| $N_{15}P_{15}K_{15}(S_{10})$ 0.4 g/L | $49.7 \pm 28.51$ [b] | $48.9 \pm 15.48$ [b] | 49.3 |
| Factor average A $LSD_{05}$ a = $F_e < F_t$ | 40.0 | 47.8 | |
| | $LSD_{05}$ ab = 41.0 | | |

The least significant difference $p < 0.05$ was calculated by two-way variance analysis; * the results were expressed as mean ± standard deviation (SD); ** "a, b, ab"—the difference between the average with the control was significant, based on the comparison of the differences between the average with LSD at a 5% significance level: "a"—by factor "a" (illumination), "b"—by factor "b" (type of fertilizer), "ab"—in the combination of factors; *** "$F_e < F_t$"–F empirical < F theoretical, not proven difference between averages with LSD at 5% significance level.

During the fourth recording, on the 56th day of observations of highbush blueberry (*Vaccinium corymbosum* L.) container-grown plants, the predominance of the previously revealed variants under LED growlights (UnionPowerStar—40W-T) treated with $N_{16}P_{16}K_{16}$ at the concentration of 0.4 g/L, $N_{15}P_{15}K_{15}(S_{10})$ at the concentration of 0.2 g/L, and $N_{16}P_{16}K_{16}$ at the concentration of 0.2 g/L was retained. However, this time no influence of the illumination (factor ("a") on the number of shoots of the 0th, 1st orders were recorded. The type of fertilizer (factor "b") and both factors in combination ("ab") significantly affected the total length of shoots (47.2 ± 8.04 and 69.0 ± 11.60 cm versus 29.3 ± 5.28 cm in the control) and the leaf surface area (127.1 ± 41.85 and 191.6 ± 50.80 cm$^2$ versus 56.1 ± 22.50 cm$^2$ in the control).

Under natural light, the predominance of the $N_{15}P_{15}K_{15}(S_{10})$-treated variant (concentration of 0.4 g/L) was revealed, where a number of the 1st order shoots were 7.6 ± 2.19 pcs. compared to 1.0 ± 0.00 pcs. for the control. Also, a significant effect of the treatment (fertilizer) type (factor "b") and both factors in combination ("ab") on the total length of shoots (40.5 ± 15.99 and 67.9 ± 16.98 cm versus 19.0 ± 7.84 cm in the control) and a leaf surface area (61.1 ± 25.53 and 128.5 ± 22.55 cm$^2$ versus 11.2 ± 7.05 cm$^2$ in the control) was shown (Table 4, Figure 3).

**Table 4.** Morphometric indicators of ex vitro development in highbush blueberry (*Vaccinium corymbosum* L.) plants cv. Brigitta Blue on the 56th day of growing completion.

| Type of Fertilizer (Factor B) | Illumination (Factor A) | | Factor Average B |
|---|---|---|---|
| | Natural Lighting + SD * | Light-Emitting Diode (LED) + SD | |
| Average number of the 0th order, pcs. | | | $LSD_{05}$ b = $F_e < F_t$ *** |
| no fertilizers (control) | 3.0 ± 1.30 | 2.6 ± 0.55 | 2.8 |
| $N_{16}P_{16}K_{16}$ 0.2 g/L | 2.8 ± 0.84 | 2.6 ± 0.55 | 2.7 |
| $N_{16}P_{16}K_{16}$ 0.4 g/L | 3.0 ± 0.89 | 3.2 ± 0.84 | 3.1 |
| $N_{15}P_{15}K_{15}(S_{10})$ 0.2 g/L | 2.8 ± 0.55 | 3.2 ± 1.22 | 3.0 |
| $N_{15}P_{15}K_{15}(S_{10})$ 0.4 g/L | 3.0 ± 0.71 | 3.0 ± 1.00 | 3.0 |
| Factor average A $LSD_{05}$ a = $F_e < F_t$ | 2.9 | 2.9 | |
| $LSD_{05}$ ab = $F_e < F_t$ | | | |
| Average number of the 1st order, pcs. | | | $LSD_{05}$ b = 2.9 |
| no fertilizers (control) | 1.0 ± 0.00 | 2.0 ± 0.71 | 1.5 |
| $N_{16}P_{16}K_{16}$ 0.2 g/L | 4.6 ± 1.52 | 2.6 ± 1.14 | 3.6 |
| $N_{16}P_{16}K_{16}$ 0.4 g/L | 3.8 ± 1.64 | 4.2 ± 1.48 | 4.0 |
| $N_{15}P_{15}K_{15}(S_{10})$ 0.2 g/L | 4.8 ± 5.56 | 3.4 ± 2.70 | 4.1 |
| $N_{15}P_{15}K_{15}(S_{10})$ 0.4 g/L | 7.6 ± 2.19 [b] ** | 3.0 ± 2.00 | 5.3 |
| Factor average A $LSD_{05}$ [a] = $F_e < F_t$ | 4.4 | 3.0 | |
| $LSD_{05}$ ab = $F_e < F_t$ | | | |
| The total length of the shoots, cm | | | $LSD_{05}$ b = 16.5 |
| no fertilizers (control) | 19.0 ± 7.84 | 29.3 ± 5.28 | 24.2 |
| $N_{16}P_{16}K_{16}$ 0.2 g/L | 61.8 ± 15.73 [b, ab] | 47.8 ± 18.92 [b] | 54.8 |
| $N_{16}P_{16}K_{16}$ 0.4 g/L | 58.1 ± 14.91 [b, ab] | 69.0 ± 11.60 [b, ab] | 63.6 |
| $N_{15}P_{15}K_{15}(S_{10})$ 0.2 g/L | 40.5 ± 15.99 [b] | 56.9 ± 7.09 [b, ab] | 48.7 |
| $N_{15}P_{15}K_{15}(S_{10})$ 0.4 g/L | 67.9 ± 16.98 [b, ab] | 47.2 ± 8.04 [b] | 57.6 |
| Factor average A $LSD_{05}$ [a] = $F_e < F_t$ | 49.5 | 50.0 | |
| $LSD_{05}$ ab = 27.5 | | | |
| Leaf surface area, cm$^2$ | | | $LSD_{05}$ [b] = 55.5 |
| no fertilizers (control) | 11.2 ± 7.05 | 56.1 ± 22.50 | 33.7 |
| $N_{16}P_{16}K_{16}$ 0.2 g/L | 128.5 ± 22.55 [a, b] | 151.8 ± 65.76 [a, b] | 140.2 |

**Table 4.** *Cont.*

| Type of Fertilizer (Factor B) | Illumination (Factor A) | | Factor Average B |
| --- | --- | --- | --- |
| | Natural Lighting + SD * | Light-Emitting Diode (LED) + SD | |
| $N_{16}P_{16}K_{16}$ 0.4 g/L | 122.8 ± 80.72 [a, b] | 191.6 ± 50.80 [a, b] | 157.2 |
| $N_{15}P_{15}K_{15}(S_{10})$ 0.2 g/L | 61.1 ± 25.53 [a] | 135.5 ± 6.63 [a, b] | 98.3 |
| $N_{15}P_{15}K_{15}(S_{10})$ 0.4 g/L | 103.0 ± 48.39 [a, b] | 127.1 ± 41.85 [a, b] | 115.1 |
| Factor average A | 85.3 | 132.4 | |
| LSD$_{05}$ [a] = 25.5 | | | |
| | LSD$_{05}$ [ab] = $F_e < F_t$ | | |

The least significant difference $p < 0.05$ was calculated by two-way variance analysis; * the results were expressed as mean ± standard deviation (SD); ** "a, b, ab"—the difference between the average with the control was significant based on the comparison of the differences between the average with LSD at a 5% significance level: "a"—by factor "a" (illumination), "b"—by factor "b" (type of fertilizer), "ab"—in the combination of factors; *** "$F_e < F_t$"–F empirical < F theoretical, not proven difference between averages with LSD at 5% significance level.

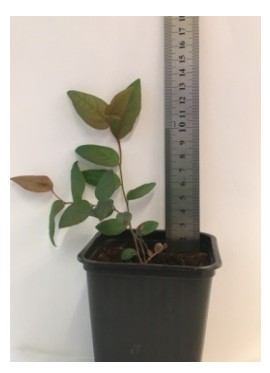 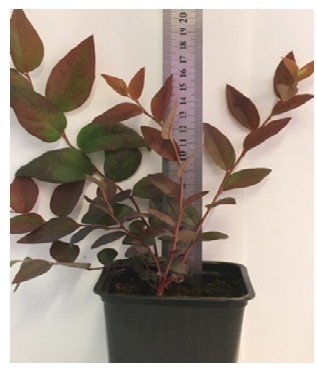 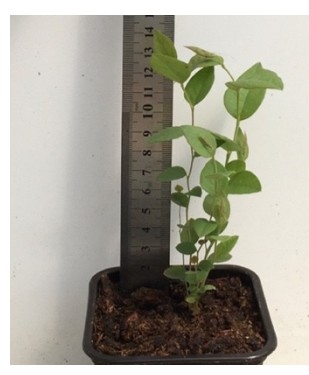 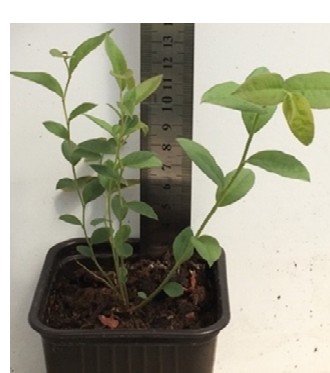

no fertilizers (control)     $N_{16}P_{16}K_{16}$ 0.4 g/L     no fertilizers (control)     $N_{16}P_{16}K_{16}$ 0.4 g/L

Light-emitting diode (LED)         Natural lighting

**Figure 3.** External appearance of ex vitro highbush blueberry (*Vaccinium corymbosum* L.) plants on the 56th day of growing completion.

On the 56th day, the growing of the highbush blueberry (*Vaccinium corymbosum* L.) was completed. Using the N-tester SPAD 502 Plus Chlorophyll Meter, we determined the indices of relative chlorophyll content in the experimental blueberry plants, since the level of chlorophyll content indicates the degree of plant maturation and plant health.

The most consistent results were obtained in variants under LED growlights (Union PowerStar—40W-T) and treated with fertilizers $N_{16}P_{16}K_{16}$ at the concentration of 0.2 g/L, $N_{15}P_{15}K_{15}(S_{10})$ and of 0.2 and 0.4 g/L, where the indices of relative chlorophyll contents were 378.3 ± 35.91 and 452.7 ± 27.93 compared to 333.7 ± 12.68 in the control.

In natural lighting conditions, the studied characteristics of experimental plants in general were generally less pronounced than in the variants under grow lights However, significant differences with the controls were obtained in all experimental variants and relative chlorophyll content indices were 331.0 ± 46.07 and 426.0 ± 25.96 compared to 273.3 ± 10.14 in the control (Table 5).

After the fourth recording on the 56th day of growing completion ex vitro, the plants were transferred to 2 L pots according to the variants, and after the feeding area had been increased, the plants that were grown under natural light demonstrated the best vigor and growth.

**Table 5.** Indices of chlorophyll content in the leaves in highbush blueberry (*Vaccinium corymbosum* L.) cv. Brigitta Blue on the 56th day of growing completion.

| Type of Fertilizer (Factor B) | Illumination (Factor A) | | Factor average B |
|---|---|---|---|
| | Natural Lighting + SD * | Light-Emitting Diode (LED) + SD | LSD$_{05}$ [b] = 59.4 |
| no fertilizers (control) | 273.3 ± 10.14 | 333.7 ± 12.68 | 353.5 |
| $N_{16}P_{16}K_{16}$ 0.2 g/L | 331.0 ± 46.07 [a] ** | 414.3 ± 38.31 [a, b] | 372.6 |
| $N_{16}P_{16}K_{16}$ 0.4 g/L | 391.3 ± 30.92 [a, b, ab] | 343.7 ± 4.71 | 367.5 |
| $N_{15}P_{15}K_{15}(S_{10})$ 0.2 g/L | 426.0 ± 25.96 [a, b, ab] | 378.3 ± 35.91 [a] | 402.1 |
| $N_{15}P_{15}K_{15}(S_{10})$ 0.4 g/L | 365.3 ± 20.42 [a, b] | 452.7 ± 27.93 [a, b, ab] | 409.0 |
| Factor average A LSD$_{05}$ [a] = 26.8 | 357.4 | 384.5 | |
| | LSD$_{05}$ [ab] = 100.0 | | |

The least significant difference $p < 0.05$ was calculated by two-way variance analysis; * the results were expressed as mean ± standard deviation (SD); ** "a, b, ab"—the difference between the average with the control was significant based on the comparison of the differences between the average with LSD at a 5% significance level: "a"—by factor "a" (illumination), "b"—by factor "b" (type of fertilizer), "ab"—in the combination of factors.

On the 112th day of growing completion under natural light, significant differences between the control and variants treated with $N_{16}P_{16}K_{16}$ (concentration of 0.4 g/L) and with $N_{15}P_{15}K_{15}(S_{10})$ (concentrations of 0.2 and 0.4 g/L), where the average number of the 0th order was 7.2 ± 2.77 and 10.2 ± 1.64 pcs. compared with 5.0 ± 2.35 pcs. in the control. As for the 1st order shoots and the total length of shoots, significant differences with the control were found in all experimental variants, regardless of the type and concentration of fertilizers. For example, the experimental plants were 1.6–2.1 times superior to the controls in terms of the total length of the shoots. As for the leaf surface area, only in the $N_{16}P_{16}K_{16}$ variant at the concentration of 0.4 g/L, was a significant effect of the fertilizer type revealed (factor "b"), this indicator was 433.7 ± 131.71 cm$^2$ as opposed to 255.1 ± 93.93 cm$^2$ in the control.

When growing under LED growlights, a significant advantage of the $N_{16}P_{16}K_{16}$ variant (concentration of 0.4 g/L) regarding the number of the 1st order shoots of (5.8 ± 2.05 pcs. versus 3.6 ± 0.55 pcs. in the control), and the total length of shoots (121.4 ± 14.31 cm versus 77.5 ± 23.90 cm in the control) was revealed. No significant effect of the fertilizer type on the total leaf surface area was found (Table 6, Figure 4).

**Table 6.** Morphometric indicators of ex vitro development in highbush blueberry (*Vaccinium corymbosum* L.) plants cv. Brigitta Blue on the 112th day of growing completion.

| Type of Fertilizer (Factor B) | Illumination (Factor A) | | Factor Average B |
|---|---|---|---|
| | Natural Lighting + SD * | Light-Emitting Diode (LED) + SD | |
| Average number of the 0th order, pcs. | | | LSD$_{05}$ b = 2.4 *** |
| no fertilizers (control) | 5.0 ± 2.35 | 3.6 ± 1.52 | 4.3 |
| $N_{16}P_{16}K_{16}$ 0.2 g/L | 5.8 ± 0.84 | 5.0 ± 1.22 [a] | 5.4 |
| $N_{16}P_{16}K_{16}$ 0.4 g/L | 8.4 ± 2.30 [a, b] ** | 4.2 ± 1.30 | 6.3 |
| $N_{15}P_{15}K_{15}(S_{10})$ 0.2 g/L | 7.2 ± 2.77 [a] | 5.0 ± 2.55 | 6.1 |
| $N_{15}P_{15}K_{15}(S_{10})$ 0.4 g/L | 10.2 ± 1.64 [a, b] | 6.4 ± 1.14 [a, b] | 8.3 |
| Factor average A LSD$_{05}$ [a] = 1.1 | 7.3 | 4.8 | |
| | LSD$_{05}$ [ab] = $F_e < F_t$ | | |
| Average number of the 1st order, pcs. | | | LSD$_{05}$ [b] = 2.9 |
| no fertilizers (control) | 3.2 ± 1.92 | 3.6 ± 0.55 | 3.4 |
| $N_{16}P_{16}K_{16}$ 0.2 g/L | 8.6 ± 4.16 [a, b] | 4.8 ± 1.30 | 6.7 |
| $N_{16}P_{16}K_{16}$ 0.4 g/L | 8.2 ± 2.17 [a, b] | 5.8 ± 2.05 [a] | 7.0 |
| $N_{15}P_{15}K_{15}(S_{10})$ 0.2 g/L | 7.4 ± 1.82 [a, b] | 4.4 ± 1.52 | 5.9 |

**Table 6.** *Cont.*

| Type of Fertilizer (Factor B) | Illumination (Factor A) | | Factor Average B |
| --- | --- | --- | --- |
| | Natural Lighting + SD * | Light-Emitting Diode (LED) + SD | |
| $N_{15}P_{15}K_{15}(S_{10})$ 0.4 g/L | $10.6 \pm 3.58$ [b] | $4.6 \pm 1.34$ | 7.6 |
| Factor average A LSD$_{05}$ [a] = 1.3 | 7.6 | 4.6 | |
| | LSD$_{05}$ [ab] = $F_e < F_t$ | | |
| The total length of the shoots, cm | | | LSD$_{05}$ [b] = 37.2 |
| no fertilizers (control) | $97.8 \pm 35.66$ | $77.5 \pm 23.90$ | 87.7 |
| $N_{16}P_{16}K_{16}$ 0.2 g/L | $156.2 \pm 32.70$ [a, b] | $96.8 \pm 28.72$ [a] | 126.5 |
| $N_{16}P_{16}K_{16}$ 0.4 g/L | $168.8 \pm 29.36$ [a, b, ab] | $121.4 \pm 14.31$ [a, b] | 145.1 |
| $N_{15}P_{15}K_{15}(S_{10})$ 0.2 g/L | $164.7 \pm 51.37$ [a, b, ab] | $109.7 \pm 24.82$ [a] | 137.2 |
| $N_{15}P_{15}K_{15}(S_{10})$ 0.4 g/L | $202.6 \pm 22.94$ [a, b, ab] | $94.2 \pm 8.94$ | 148.4 |
| Factor average A LSD$_{05}$ [a] = 17.1 | 158.0 | 99.9 | |
| | LSD$_{05}$ [ab] = 61.9 | | |
| Leaf surface area, cm$^2$ | | | LSD$_{05}$ [b] = 116.2 |
| No fertilizers (control) | $255.1 \pm 93.93$ | $280.9 \pm 61.63$ | 268.0 |
| $N_{16}P_{16}K_{16}$ 0.2 g/L | $323.3 \pm 37.13$ | $305.9 \pm 120.04$ | 314.6 |
| $N_{16}P_{16}K_{16}$ 0.4 g/L | $433.7 \pm 131.71$ [b] | $305.4 \pm 138.92$ | 369.6 |
| $N_{15}P_{15}K_{15}(S_{10})$ 0.2 g/L | $370.4 \pm 97.23$ | $243.4 \pm 59.52$ | 306.9 |
| $N_{15}P_{15}K_{15}(S_{10})$ 0.4 g/L | $194.2 \pm 55.28$ | $220.2 \pm 57.28$ | 207.2 |
| Factor average A LSD$_{05}$ [a] = $F_e < F_t$ | 315.3 | 271.2 | |
| | LSD$_{05}$ [ab] = $F_e < F_t$ | | |

The least significant difference $p < 0.05$ was calculated by two-way variance analysis; * the results were expressed as mean $\pm$ standard deviation (SD); ** "a, b, ab"—the difference between the average with the control was significant based on the comparison of the differences between the average with LSD at a 5% significance level: "a"—by factor "a" (illumination), "b"—by factor "b" (type of fertilizer), "ab"—in both factors combined; *** «$F_e < F_t$»–F empirical < F theoretical, not proven difference between averages with LSD at 5% significance level.

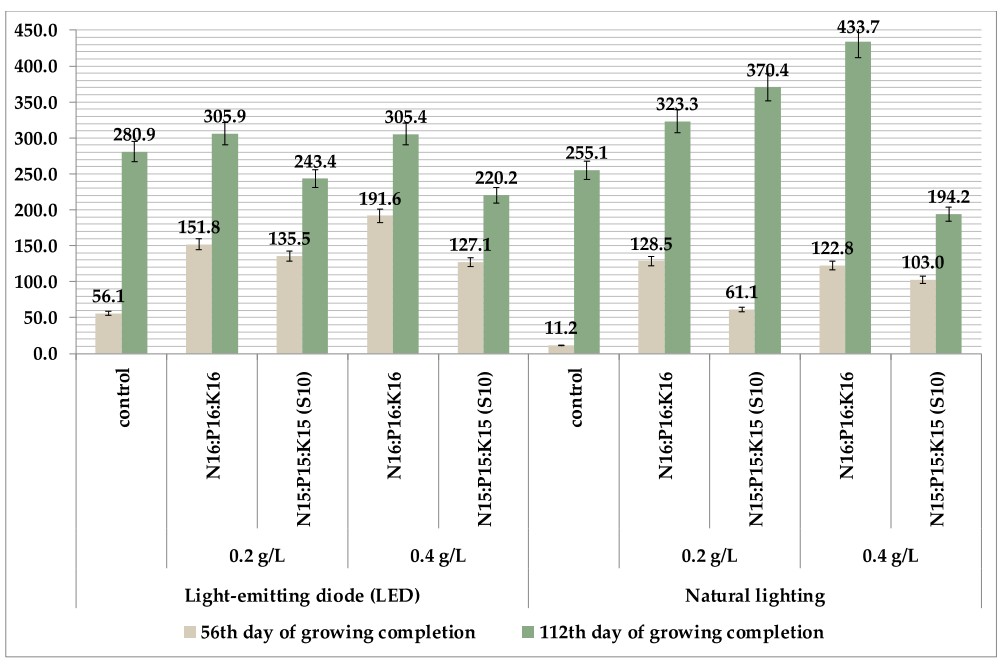

**Figure 4.** *Cont.*

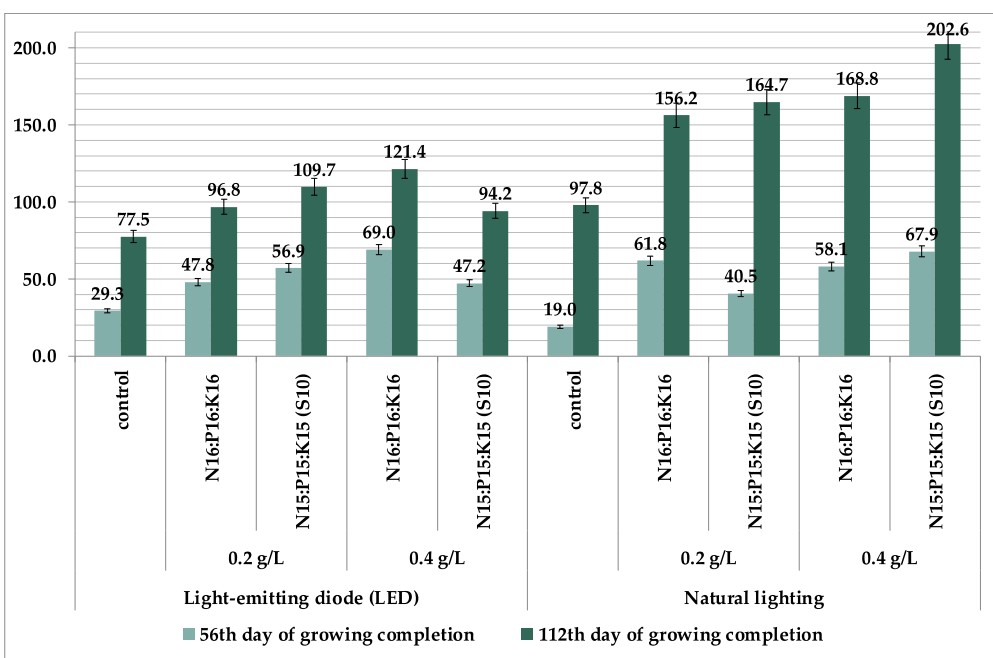

Total of the leaf surface area, cm²

**Figure 4.** Ex vitro development indexes of highbush blueberry (*Vaccinium corymbosum* L.) plants on the 56th and 112th days of growing completion.

In general, the LED growlights appear to have a positive effect on the development of container-grown plants only at the initial stages of growing completion ex vitro (up to the 56th day of cultivation). After transplanting into larger pots, a significant change in the dynamics of development morphometric indicators was revealed under the natural light. On the 112th day of growing in the variant treated with $N_{16}P_{16}K_{16}$ with the concentration of 0.4 g/L, the total shoot length and the leaf surface area were 1.4 times higher than those of plants grown under growlights (Figure 4).

## 4. Discussion

Our study is focused on the biogenesis of mature blueberry plants in the open field to provide high crop productivity. However, the influence of individual macro- and microelements on the adaptive potential of fruit and berry plants is not yet completely understood. Outdated planting techniques negatively affect the plant growth and development and do not allow full realization of the crop's biological potential and productivity.

It will be possible to overcome limitations and ensure high productivity of natural (in the forest) and commercial plantations of highbush blueberry (*Vaccinium corymbosum* L.) thanks to the transcendental capabilities of Biogeosystem Technique (BGT*) [65,66], which provides new opportunities in the management of soil structure and architecture and can solve problems related to soil moisture [67], organic matter content [68] and HM [3], recultivation of fertilizing, structure-forming and stimulating substances in the soil [69], including nanomaterials [41,70,71].

Additional lighting in the cultivation conditions of facilities has been widely used for more than a century to accelerate plant growth and development [72–75].

Over the past few decades, the light-emitting diode (LED) technology has been increasingly used for lighting in horticulture due to its cost-effectiveness and ease of use [50,74]. Moreover, the use of such lighting devices is popular due to their longer service period than that of fluorescent lamps, which have a fairly short life [76,77].

In fact, the success of ex vitro regeneration of plants is determined by their ability to produce new shoots that can adapt to new cultivation conditions. Lighting under ex vitro conditions plays an important role in the development of the plant photosynthetic capacity.

It was found that using LEDs as a light source during the development of microshoots in vitro, as well as in subsequent stages of adaptation and post-adaptation, increases the rate of plant development [22,39,40].

Blue light has been found to enhance vegetative bud initiation in in vitro conditions [78]. For half-highbush blueberry at the in vitro rooting stage, the addition of blue and red light spectrums promoted more intensive root formation seven times [79]. This suggests the prospect of using this illumination in the subsequent ex vitro development of plants. In addition, the presence of blue light in the spectrum of plant illumination is a key factor influencing the stomatal apparatus [80,81]. Plants after in vitro culture during the period of adaptation to non-sterile environment conditions and subsequently of growing completion, need to restore the working of stomata, therefore blue must be included in the lighting spectrum.

Numerous studies have confirmed that a spectral combination of red and blue light in various ratios is quite effective for growing various plants in greenhouse conditions [47]. Moreover, the plants' reaction to the light spectrum differs [48]. The blue light is effective for inducing and accumulating a chlorophyll content, a carbon–nitrogen ratio (C/N ratio) and the plastid number in the guard cells in ex vitro *Actinidia* plants, while the red light induces a vegetative plant growth [51].

It is known that the ex vitro highbush blueberry plants grown in protected cultivation in containers after the stage of adaptation to non-sterile conditions are characterized by very slow growth and development [62]. This can be caused by various reasons, such as lack of nutrients, insufficient light levels, wrong selection of the light spectrum and temperature conditions.

Our studies have confirmed the high efficiency of phyto-illumination with LED grow-lights (UnionPowerStar—40W-T) with a photoperiod of 16/8 h at the initial ex vitro growing stages (until the 56th day of cultivation) of highbush blueberry (*Vaccinium corymbosum* L.) cv. Brigitta Blue in low-volume containers (0.5-L).

In addition, it is known that the growth and development of ex vitro plants in containers is influenced by the concentration and ratio of nutrients [28]. When growing ex vitro plants, it is important to ensure a uniform supply of macro- and micronutrients in as low concentrations as possible so that the roots of young plants are compatible with the nutrient absorption rate depending on the volume of substrate in the container, and to maintain acidity of the soil substrate [82,83].

Excessive amounts of mineral fertilizers slow down the growth of blueberry plants. Development rate and winter hardiness of an aboveground system decreases. High doses of nitrogen can lead to excessive growth of vegetative mass (canopy), increasing the growing season, which can affect the condition of the planting material [7,84]. When growing ex vitro plants in containers, it is also necessary to consider the sensitivity of highbush blueberry (*Vaccinium corymbosum* L.) seedlings to low-volume nutrition of the root system and not to allow increased phosphorus content, because at higher values it prevents iron absorption. At the same time, it is possible to control the root system by the size of container, as a smaller container can limit the growth of the above-ground system slowing down the development of the root system [45,46,70,85].

High doses of fertilizers can increase chlorophyll amount in leaves [84]. It has been found that 50 to 70% of nitrogen in leaves is associated with enzymes present in chloroplasts, indicating a direct correlation between nitrogen and chlorophyll content [86,87]. In addition, the chlorophyll content is crucial for the efficiency of photosynthesis and therefore adaptation to different conditions and obtaining better quality seedlings [87].

In our studies, it was found that on the 56th day of growing completion, in all variants of the experiment, regardless of the light source, fertilizer type and concentrations (with the exception of $N_{16}P_{16}K_{16}$ at the concentration of 0.4 g/L with LED) significant differences with the controls in terms of indices of relative chlorophyll content were revealed. In the variants exposed to LED growlights, the best consistent results were obtained when using $N_{16}P_{16}K_{16}$ (0.2 g/L), $N_{15}P_{15}K_{15}(S_{10})$ (0.4 g/L), where the relative chlorophyll content

indices were 414.3 ± 3 8.31 and 452.7 ± 27.93 compared to 333.7±12.68 in the controls; in variants with natural light, when using fertilizer $N_{15}P_{15}K_{15}(S_{10})$ (0.2 g/L), the indices of relative chlorophyll content were 426.0 ± 25.96 compared to 273.3 ± 10.14 in the controls.

Doses of mineral fertilizers for ex vitro blueberry plants depend on the substrate used, the phase of plant development, and the type of fertilizer [88]. The highest effect is observed when using complex mineral fertilizers, and they can have a positive effect on the growth and development of both seedlings and fruit-bearing blueberry plants [7,89].

As a result of our research, it was revealed that LED growlights have a positive effect on the development of ex vitro plants of highbush blueberry (*Vaccinium corymbosum* L.) cv. Brigitta Blue only at the initial stages of growing in 0.5-L containers. Until the 56th day of growing, it is effective to use LED grow lights as a source of illumination and apply mineral fertilizers, such as $N_{16}P_{16}K_{16}$ (0.2 and 0.4 g/L) and $N_{15}P_{15}K_{15}(S_{10})$ (0.2 g/L) to the substrate.

Upon transplanting into 2 L pots, the advantage of developing experimental ex vitro plants in natural light conditions was revealed. On the 112th day of growing completion, significant predominance of the variant treated with the fertilizer $N_{16}P_{16}K_{16}$ (0.4 g/L) was revealed using the indicators of the total shoot length and the leaf surface area which were 1.4 times higher than those of plants grown under growlights.

## 5. Conclusions

In our research, we studied ex vitro cultivation methods of highbush blueberry (*Vaccinium corymbosum* L.) cv. Brigitta Blue in greenhouse conditions. Two types of lighting were used (LED lighting with a 16-h photoperiod and natural lighting—without additional illumination tools). At the same time, ex vitro plants were planted in a peat substrate and mineral fertilizers were applied. In containers of 0.5 L, until the 56th day of development, experimental plants grew better under LED lighting conditions with the application of mineral fertilizers $N_{16}P_{16}K_{16}$ (0.2 and 0.4 g/L) and $N_{15}P1_5K_{15}(S_{10})$ (0.2 g/L).

After transplanting into larger 2 L pots, a significant change in the plant development dynamics and the prevalence in morphometric indicators of ex vitro plants grown under natural light conditions were revealed. On the 112th day of the experiment, in the experimental variant of substrate with mineral fertilizers $N_{16}P_{16}K_{16}$ (0.4 g/L) the indices of the total shoot length and the leaf surface area were 1.4 times higher than those of the plants growing under LED grown lights.

These results obtained have improved our view of lighting and mineral fertilizers and their effect on the development of ex vitro plants of highbush blueberry (*Vaccinium corymbosum* L.) cv. Brigitta Blue in greenhouse conditions. They can be useful for providing blueberry planting stock and commercial use for large scale production.

An improved highbush blueberry organogenesis ex vitro and HM free production in situ will be provided via the BGT* methodology of intrasoil milling, intrasoil pulse continuously-discrete watering and intrasoil dispersed matter application during intrasoil milling.

**Author Contributions:** Conceptualization S.A.; Methodology S.A., E.S. and A.R.; formal analysis A.Z. and T.A.; investigation E.S., I.N. and P.K.; resources Y.V. and A.M.; data curation S.A., I.N. and P.K.; writing—original draft preparation E.S., I.N. and P.K.; writing—review and editing S.A., A.R. and A.E.; visualization A.Z., I.N. and P.K.; supervision Y.V., T.A. and A.M.; project administration S.A., A.R. and A.E.; funding acquisition S.A., A.R. and A.E. All authors have read and agreed to the published version of the manuscript.

**Funding:** This research was funded by the program of development of the Russian State Agrarian University—Moscow Timiryazev Agricultural Academy "Agroproyv-2030" during the strategic academic leadership program "Priority-2030" (No. 075-15-2021-1160).

**Institutional Review Board Statement:** Not applicable.

**Informed Consent Statement:** Not applicable.

**Data Availability Statement:** Not available.

**Conflicts of Interest:** The authors declare no conflict of interest.

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
