# Peer review of "Improvement of Ex Vitro Growing Completion of Highbush Blueberry (Vaccinium corymbosum L.) in Containers"

_forests, doi:10.3390/f13101550_

Round 1
Reviewer 1 Report
The manuscript titled „IMPROVEMENT OF EX VITRO COMPLETING OF GROWING OF HIGHBUSH BLUEBERRY (VACCINIUM CORYMBO- SUM L.) IN CONTAINERS” is a great paper about the topic. Today, there is a keen interests in blueberry production, there are lots of innovation in this indusrty to spread its area worldwide.
The examined cultivar was ’Brigitta Blue’ in this research. Please add a short description with phenological data about this cultivar to give a change to know it for those readers, who don’t have any information about it. Please replace „variety” to „cultivar” in the text.
I think it is also important to describe the soil or media conditions put into the containers, where the plants were planted in. The special plant-care activities were described in the manuscript, but the general plant-care activities (e.g. irrigation, increasing the air humulity content etc.) are missing. When or how many times were the number of shoots, the total lenght of shoots, and the total leaf surface area counted? Please add more detailed information to these measurements. I think this is one of the most important issues of this paper.
Reviewer 2 Report
The aim of this study was to determine the effects of light and fertilization on the ex vitro growth and development of V. corymbosum 'Brigitta Blue'. Applied methods are adequate but very simple and do not introduce new techniques into the field of research. The growth and development of wild V. corymbosum and cultivars cannot be generalized. The research material is cv. 'Brigitta Blue' developed in Australia, recommended for cultivation in plantations and gardens, but not in the forest. The negative evaluation is due to numerous objections:
1. Abstract should not contain information off topic (Line 12-14, Line 24-26)
2. The Introduction does not provide a clear statement of the problem, the relevant literature on the subject, and proposed approach or solution.
3. Some statements are not clear and do not relate to the experiments, for example such as Line 34-36.
4. The aim of study is not clearly formulated. What does „protected ground conditions” mean?
5. There are not enough details presented in the “Materials and methods. There is no information about the plant material, by what method it was propagated in vitro, rooted and acclimatized. It is not clear where the experiment was performed, what was PPFD under LED lighting? The control plants were grown in natural light, but there is no information about the length of the day and daily light integral. What was the spectral quantification and temperature during experiment? It is not clear what the „turfy substrate” is? The names of fertilizers and their manufacturers are missing.
6. Result chapter. The results would be more transparent if they were shown in graphs, and showed the effect of the factors studied on the trait at different times, for example, plant height of V. corymbosum ‘Brigitta Blue’ after 14, 28, 42 and 56 days of growth under …..
The question is whether the positive effect of LED light was due to its spectrum or the length of the day at the beginning of plant growth?
7. Discussion. Large sections of the discussion are off topic, for example Line 303-320, Line 321-336 (experimental material was already acclimatized). The effect of light and fertilization has been previously studied by many autors, including Petridis et al. 2018, Gomez et al. 2012, Hung et al. 2016, Kim et al. 2011, Nacheva et al. 2022, Escobar et al. 2017, Petridis et al. 2018.
8. Conclusions (Line 412-429) are poorly formulated, they are a repetition of the results
Last paragraph is unclear, seems off topic.
Additional comments:
English name of plant species should be written small letter. The name of cultivar should be not in italics. The name of cultivar should be denoted by single quotes, but not after the word "variety" or abbreviation “cv.”; 'Brigitta Blue', but cv. Brigitta Blue.
Round 2
Reviewer 2 Report
Changes made by the Authors have improved the work to some extent. However, the work still has many shortcomings that affect my negative assessment.
1. Still many statements are off topic There is no link between the experiments conducted and the forest ecosystem or Biogeosystem Technique (please see attached file).
2. It is not clear what was the natural day lenght for control plants, without any supplemental lighting? Light was generated by blue and red diodes (line 178-179), but was the range of red to blue light? What was the spectral quantification for natural lighting and light emitting diodes? Please answer the question whether the positive effect of LED light was due to its spectrum or the length of the day at the beginning of plant growth? There is not clear why Authors use so low PPFD.
3. The light experiment is poorly disscused.
